# Research Progress on Heavy Metal Passivators and Passivation Mechanisms of Organic Solid Waste Compost: A Review

Yuanping Zhong [1,2], Wenqing Yang [2,3,*], Qian Zhuo [3], Zhi Cao [4], Qinghua Chen [2] and Liren Xiao [1,*]

1 College of Chemistry and Materials Science, Fujian Normal University, Fuzhou 350007, China; m15860227386@163.com
2 Fujian Key Laboratory of Pollution Control & Resource Reuse, Fujian Normal University, Fuzhou 350007, China
3 Fujian Provincial Key Lab of Coastal Basin Environment, Fujian Polytechnic Normal University, Fuqing 350300, China
4 Fujian Provincial Engineering Research Center of Modern Facility Agriculture, Fujian Polytechnic Normal University, Fuqing 350300, China
* Correspondence: yangwq@fpnu.edu.cn (W.Y.); xlr1966@126.com (L.X.)

**Abstract:** Organic solid waste is a renewable resource as it can be transformed into a valuable product through various technologies. Composting is considered to be the most economical and effective technology for treating organic solid waste, but excessive amounts of heavy metals in organic solid waste compost are harmful to the environment. The current focus is on the addition of heavy metal passivators to organic solid waste to reduce the mobility and biotoxicity of heavy metals in situ or ex situ. The aims of this paper are to provide an overview of heavy metal passivators and their passivation mechanisms in the field of organic solid waste composting and to provide a reference for research on the control of heavy metal pollution in the treatment of organic solid waste.

**Keywords:** organic solid waste; composting; heavy metals; passivators; passivation mechanism

## 1. Introduction

With the continuous growth of the global population, accelerated urbanization, and the increased scale of livestock and poultry farming, a large amount of solid waste is being generated [1]. According to Hoornweg et al., China's solid waste generation is expected to increase from 520,550 tons per day in 2005 to 1.4 million tons per day in 2025 [2]. Organic solid waste is defined as material that has lost its original value, has a moisture content of less than 85%–90%, and is biochemically degradable; this accounts for over 40% of the total solid waste in China. Organic waste includes kitchen waste, farmland and garden waste, livestock manure, municipal waste, and sludge [3]. According to the China Statistical Yearbook and other relevant data, in 2021, China's kitchen waste production was approximately 127 million tons; the national urban municipal sludge production was approximately 55.52 million tons per year; the national livestock and poultry manure production was 3.05 billion tons per year; and the gardening waste production reached 5 million tons a year in Beijing alone. Therefore, the efficient use of organic solid waste has become one of the most pressing issues in China and is very important for both social production and environmental sustainability [4].

Common organic solid waste treatment methods include landfill, incineration, pyrolysis, gasification, and composting [4]. In particular, composting has received much attention as an economical, environmentally friendly, and sustainable treatment method [5]. Composting is a biochemical process that transforms organic solid waste into stable humus (HS) through the action of microorganisms [6]. Composting can reduce the volume of organic solid waste and can be used in crops after maturation not only to provide organic nutrients but also to destroy weed seeds and pathogens to a certain extent [7]. However,

excessive heavy metals in the composting process are still a serious problem. For example, most of the heavy metal additives added to livestock and poultry feed cannot be absorbed by the intestinal tract of livestock and poultry and will be excreted with feces, posing a great threat to the environment [8]. Although organic solid waste in the form of compost can passivate some of the heavy metals, the mobile fraction of heavy metals remains high and can easily enter the food chain through bioconcentration after entering the soil, causing harm and affecting the prospect of its effective recycling.

Currently, the addition of heavy metal passivators is considered to be the most economical and effective method for the passivation of heavy metals in compost [9,10]. The passivation of heavy metals in compost is influenced by a number of factors such as the physicochemical, microbiological, and organic composition [11]. The addition of heavy metal passivators can alter the physicochemical properties of the compost and affect the structure of the microbial community; it can also promote the formation of humic acid (HA) that is closely related to the morphological transformation of heavy metals in the composting process and can form complexes with heavy metals [12,13]. It has also been shown that heavy metal compost passivators such as phosphates can promote the formation of residue states in a co-precipitating manner to passivate heavy metals [14]. However, no relevant studies have focused on the overall and recent developments in the classification and passivation mechanisms of compost heavy metal passivators. Therefore, a summary analysis of the scientific work related to the heavy metal passivation of organic solid waste compost is necessary and urgently needed.

This article presents a comparison of the effects and mechanisms of different heavy metal passivators in the organic solid waste composting process. This article briefly introduces the treatment of heavy metals in organic solid waste and describes the definition, classification, and application of heavy metal passivators for composting. Finally, we focus on the mechanisms of heavy metal passivation in organic solid waste composting, including ion exchange conversion, microbial community impact, humification promotion, and biomineralization migration. The objectives of this review are to provide a timely and comprehensive understanding of the roles of various composting passivators in the treatment of organic waste, to identify shortcomings and limitations in current research, and to clarify future research directions and prospects for the use of heavy metal passivators. The X-MOL Academic Platform and the Web of Science Core Collection serve as the primary literature data sources for this study while incorporating a comprehensive collection of compost research materials accumulated by the research group over the years as supplementary references.

## 2. Technologies for the Treatment of Heavy Metals in Organic Solid Waste

In recent years, due to changes in human lifestyles, the diversification of the composition of organic solid waste has been intensifying, and the possibility of contamination by heavy metals has greatly increased. For example, the excessive addition of heavy metal additives to livestock feeds has led to high levels of heavy metals in livestock manure that, if left untreated, can pose serious threats to the ecological environment as well as to human health [15]. Currently, common technologies for the treatment of heavy metals in organic solid waste include pyrolysis, landfill immobilization, construction material immobilization, and composting.

Pyrolysis is a heavy metal treatment technology that uses high temperatures to convert some heavy metals in a bioavailable state into a more stable form. The ecotoxicity of heavy metals is significantly reduced after pyrolysis [16,17]. For example, Xie et al. [18] used industrial sludge co-pyrolysis with rice straw (RS) at different temperatures and ratios to find the best potential for application of the char obtained at 600 °C, and a 20% addition of RS promoted the conversion of Cr, Zn, and Cd from a bioavailable state to more stable forms with lower leaching toxicity and potential ecological risk. The residual state fraction of Zn increased from 29.10% to 39.99%. The residual state of Cd increased from 29.99% to 75.85% as the RS ratio increased to 40% at 600 °C. Devi and Saroha [19] investigated the effect

of pyrolysis temperature on the partitioning and mobility of six heavy metals (Cr, Cu, Ni, Zn, Pb, and Cd) in pyrolysis biochar made from pulp and paper mill wastewater treatment plant sludge and found that most of the heavy metals were present in residue form in the biochar pyrolysis at 700 °C. For example, the residue state of Cr increased from 30.81% to 47.51% after pyrolysis at 700 °C.

The establishment of landfills remains one of the main ways of treating organic solid waste in developing countries, and it can be effective in preventing the leaching of heavy metals [20]. Landfill immobilization techniques for heavy metals usually rely on the formation of metal salts, complexation with fulvic acid (FA) and HA, biosorption by microbial cell walls, and chemical precipitation during the landfill process. Srivastava et al. [21] utilized dry tomb landfilling–bioreactor landfilling (DTL–BRL) and dual-mode anaerobic landfilling, combining municipal and industrial solid waste. The dual-mode landfilling operation was effective in reducing the bioavailability of heavy metals after 300 days of operation, with residue states of Cr, Ni, Pb, Cu, Cd, and Zn increasing by 4.69%, 8.45%, 6.80%, 6.84%, 3.40%, and 4.35%, respectively. Yao et al. [22] established a simulated landfill and operated it for 507 days to explore the pattern of heavy metal immobilization by municipal solid waste (MSW) during the landfill process; they found that 25 kg of MSW could immobilize 45.6 mg, 3.16 mg, and 44.2 mg of Cu, Pb, and Zn, respectively, during the entire landfill process. This mode of immobilization is largely due to the combination of heavy metals with humic substances in the landfill to form hydroxides, sulphones, and carbonates and dur to physical adsorption that reduces the mobility of the heavy metals.

Considering that many new lightweight construction materials require organic solid waste as a binder and pore-forming agent, construction material immobilization technology is considered to be a relatively simple and effective method of passivation for the sequestration of heavy metals from organic solid waste [23]. Chang et al. [24] suggested that recycling sewage sludge (SS) as a raw material in construction materials not only transforms organic solid waste into a valuable product but also immobilizes heavy metals and prevents secondary environmental pollution. Guo et al. [25] suggested that gelling materials could encapsulate heavy metals by curing and bonding solid waste particles containing heavy metals, that the mixture would harden to produce a monolithic waste with structural integrity and long-term stability, and that the waste would be isolated in a form that inhibits leaching of heavy metals and reduces the environmental risk, despite the increased mass and volume of the waste.

Composting as a microbial conversion method has clear advantages in organic solid waste treatment due to its low environmental risk and excellent resource reusability [6,26,27]. The composting process reduces the mobility and bioavailability of heavy metals by converting them from their extractable and reducible state to a more stable form [28]. Composting is therefore considered to be an effective method for passivating heavy metals in organic solid waste [29]. Compost can interact with the surface of the material through the formation of complexes or ion exchange, and metal ions can be attached to amide groups in the composted organic material for removal purposes [30]. In addition, bacteria and fungi in the composting process also play crucial roles in improving metal removal by enhancing the degradation of organic matter (OM) for humification and their own biosorption capacity. A large number of studies have demonstrated the positive effect of organic solid waste composting on passivating heavy metals. By studying the changes in heavy metals during the co-composting of rural organic solid waste (SS, kitchen waste, and maize straw), Xu et al. [31] found that the oxidizable and residual fractions of heavy metals dominated the composting process, indicating the relative stability of heavy metals after composting; they also found that, after composting, the content of heavy metals met the standards and was safe for use in agricultural applications. Azizi et al. [32] removed 90% of the heavy metals, including Cr, Cd, and Pb, by vermicomposting with spent mushrooms (SMC) and modified municipal SS. The stability of the compost product as a fertilizer for agricultural use was unaffected.

## 3. Definition, Classification, and Application of Heavy Metal Passivators in Organic Solid Waste Composting

Tessier [33], in 1979, developed an analytical procedure involving the sequential chemical extraction of particulate trace metals (Cd, Co, Cu, Ni, Pb, Zn, Fe, and Mn) into five fractions: the exchangeable state, the carbonate-bound state, the Fe-Mn oxide-bound state, the OM-bound state, and the residue state. Although more time consuming, this process provides detailed information on the source, mode of occurrence, bioavailability, physicochemical availability, migration, and transformation of trace metals. The first three forms of the Tessier classification (exchangeable, carbonate-bound, and Fe-Mn oxide-bound) are generally considered to be readily transported and transferred into the environment as bioavailable forms [34]. The latter two forms—the organic-bound states and residue states—are relatively stable and are often referred to as passivated forms.

In 1987, a series of investigations and collaborative studies initiated by the Bureau Community of Reference (BCR) using chemical reagents to extract metal species formation from soils and sediments led to the BCR three-step extraction method [35]. The elemental distribution of this method comprises exchangeable (EXC), reducible (RED), oxidizable (OXI), and residual (RES) states, and it is generally accepted that the exchangeable and reducible states of heavy metals are easily transported into the environment and are considered bioavailable heavy metals [36]. The oxidizable and residue states are correspondingly bio-unavailable passivated forms of heavy metals. The bioavailability factor (BF) is generally considered to be the ratio of the exchangeable, reducible state to the exchangeable, reducible, oxidizable, and residual states, and it is calculated as follows:

$$BF = (EXC + RED)/(EXC + RED + OXI + RES)\%, \tag{1}$$

In practice, many researchers have improved on these classical methods of heavy metal extraction and classification and have applied them [37,38]. However, the overall line of research is consistent.

Heavy metal passivation in organic solid waste composting refers to the reduction of the activity and biological effectiveness of heavy metals through physical, chemical, and biological synergists in contact with the heavy metals in organic solid waste composting through adsorption, precipitation, co-precipitation, or stabilization with heavy metal ions during the composting process [39]. By adding passivators, compost undergoes a series of changes such as in the structure of the compost microbial community, the release of active phosphorus, the degree of humification, conductivity, pH, oxidation–reduction potential (ORP), and other physicochemical or biological characteristics. At present, there is no definite standard for the classification of passivators. According to the characteristics of the passivator, the passivators used in the current research are divided into physical passivators, chemical passivators, and biological passivators.

### 3.1. Physical Type of Compost Passivator

Physical compost passivators include alkaline mineral materials and high surface area adsorbent materials such as biochar, zeolite, zero valent iron (ZVI), activated carbon, and sepiolite (SEP). Due to their large electrostatic forces, ion exchange properties, and large cavity surfaces, the biological effects of heavy metals are reduced through physical adsorption. Table 1 readily infers the passivation ability of various physical passivators.

**Table 1.** Passivation of heavy metals in organic solid waste compost by physical passivators.

| Types and Proportions of Passivators | Passivated Heavy Metals | Composting Materials | Passivation Effect | References |
|---|---|---|---|---|
| 10% Biochar | Zn, Cu, Cd, Pb | Pig manure | BF decreased by 4.10%, 44.12%, 18.75%, and 30.06%, respectively | [40] |

**Table 1.** *Cont.*

| Types and Proportions of Passivators | Passivated Heavy Metals | Composting Materials | Passivation Effect | References |
|---|---|---|---|---|
| 7% Activated carbon | Cu, Zn | Chicken manure | BF decreased by up to 84% and 10%, respectively | [41] |
| 10% Zeolite | Cu | Pig manure | BF decreased by up to 3.1% | [42] |
| 20 g/kg ZVI | Zn, Cu | Dairy manure | BF decreased by up to 8.25% and 19.84%, respectively | [43] |
| 9% Sepiolite | Cu, Zn | Pig manure | BF decreased by 33.3% and 32.7%, respectively | [44] |

Chang et al. [45] investigated the effects of adding different proportions of peanut shell biochar (PB) to the heavy metals Cu and Zn in the high-temperature composting of wheat straw SS and extracted the products using the BCR continuous extraction method. It was found that EXC-Cu and RED-Cu were the main forms of Cu and that RED-Zn was the main form of Zn in the early stage of the composting process. In the late stage of composting, the proportions of EXC-Cu and RED-Cu decreased by 9.72% while the proportion of EXC-Zn and RED-Zn decreased by 2.03% following a 30% addition of PB. Hu et al. [46] investigated the dynamics of heavy metal fractions in the aerobic composting of pig manure by adding zeolites of different particle sizes. The addition of coarse zeolites (particle size 3–5 mm) showed good passivation effects on Cu, Cd, and Pb, with the BF decreasing for Cu (45.13%), Cd (16.11%), and Pb (25.49%). The passivation of Cu and Cd was evident in the decaying stage, while the passivation of Pb was mainly in the thermophilic stage. Zhang et al. [41] investigated the dynamic changes of heavy metal fractions in the composting process of activated carbon (AC)-amended chicken manure at different periods of composting and found that the passivation of heavy metals by AC showed different priorities and different passivation patterns. The passivation of Cu was accelerated by AC through the regulation of environmental factors, and the BF of Cu was reduced by up to 84% through the influence of the microbial community. Wang et al. [47] investigated the nitrogen retention capacity and the ability to passivate heavy metals during pig manure (PM) and sawdust composting with the addition of different concentrations of mackerel (MS) and found that the addition of MS reduced the biological effectiveness of Cu and Zn and reduced $NO_3$-N production at the end of composting. Zheng et al. [44] investigated the passivation ability of Cu and Zn during the aerobic composting of PM by adding different proportions of SEP and found that the bioavailability of the heavy metals Cu and Zn in the compost decreased with increasing amounts of SEP. The BF values of Cu and Zn decreased by 33.3% and 32.7%, respectively, with 9% SEP addition compared with the initial compost mixture. This series of well-established studies has demonstrated the reliable performance and potential of physical compost passivators for the treatment of heavy metals in compost.

*3.2. Chemical Type of Compost Passivator*

Chemical compost passivators are designed to immobilize heavy metals in a stable form by means of a chemical reaction. For example, some studies have added alkaline passivators to increase the pH of the pile to reduce the biological effectiveness of heavy metals [47,48].

**Table 2.** Passivation of heavy metals in organic solid waste compost by chemical treatment.

| Types and Proportions of Passivators | Passivated Heavy Metals | Composting Materials | Passivation Effect | References |
|---|---|---|---|---|
| 2.5% Rock phosphate | Pb | Sewage sludge | BF decreased by 6.88% | [49] |

**Table 2.** *Cont.*

| Types and Proportions of Passivators | Passivated Heavy Metals | Composting Materials | Passivation Effect | References |
|---|---|---|---|---|
| 10% Calcium magnesium phosphate fertilizer | Cu | Pig manure | BF decreased by 11.47% | [50] |
| 7.5% Rock phosphate | Cu, Zn | Pig manure | BF decreased by 23.8% and 0.64%, respectively | [51] |
| 7.5% Rock phosphate | Cu | Pig manure | BF decreased by 47.24% | [52] |

The current chemical passivators are represented by calcium phosphate, calcium magnesium phosphate, rock phosphate, and calcium superphosphate fertilizers—primarily phosphorus–calcium system passivators as shown in Table 2. Liu et al. [42] investigated the effect of phosphonyls with calcium oxide (PPG + CaO) on compost maturation and heavy metal passivation in PM compost and found that PPG + CaO treatment further enhanced the transfer of Cu (65.6%), Cd (21.7%), and Pb (48.7%) to the steady state (OXI and RES), and the significant increase in humification was the main reason why PPG + CaO enhanced the passivation rate of composted heavy metals. Wang et al. [51] investigated the effect of rock phosphate on the bioavailability of Cu and Zn in the co-composting of PM and RS and found that the addition of 7.5% rock phosphate showed the best passivation effect on the bioavailability of Cu, with a 23.8% reduction in the BF, and that the cations in the passivator were exchanged for $Cu^{2+}$ and $Zn^{2+}$ on the surface of the compost material, thereby promoting co-precipitation with phosphate on the Fe-Mn oxide surface. Chemical passivators reduce the bioavailability of heavy metals in organic solid waste compost, but due to their non-recyclable nature, they can easily cause secondary pollution of the environment if the dosage and compost application are not properly controlled.

*3.3. Biological Type of Compost Passivator*

Biological compost passivation refers to the inoculation of compost with microorganisms that increase the ORP and lower the pH of the compost or to the addition of exogenous organisms to passivate compost containing heavy metals. The most prominent types of biological passivation agents are those that increase the HS content of compost by using reactive groups such as HA -COOH and phenol -OH groups that bind strongly to heavy metals and thus affect the concentration and mobility of free and unstable metal ions. Li et al. [53] investigated the effect of microbial inoculation (MI) on heavy metal (Cr, Cd, and Pb) fractions in PM compost and found that MI could facilitate the transfer of heavy metal fractions by influencing the levels of HA and FA, specifically contributing to increases in the residue fractions. Hait et al. [54] studied the transformation of some heavy metals (Cu, Co, Fe, Mn, Zn, and Cr) during SS vermicomposting and found that the system may reduce the biological effectiveness of heavy metals in order to reach the point where the SS is converted and recycled into a nutrient-rich organic fertilizer. The fertilizer and conditioner were applied in agriculture without any agroecological toxic effects, which the authors suggest may be related to the activity of earthworms and the formation of organically bound complexes during the mixed composting processes. Wei et al. [55] investigated the effect of adding HS and heavy metal resistant bacterial communities to maize straw compost on the removal of heavy metals during the composting process; their results showed that the combined addition of HS and bacteria had a significant effect on the biosorption of heavy metals, removing 60–80% of heavy metals during the composting process. This process also plays a crucial role in increasing the diversity and biomass of the bacterial community.

Physical passivators, which have a wide range of applications and sources, have a good passivation effect on heavy metals and are currently most widely added to compost. Chemical passivators, due to the great cost of raw materials compared with physical passivators, are less effective in passivating a complex variety of heavy metals due to

their more homogeneous composition. Biological passivators need to have exogenous organisms added to the compost, which requires strict composting conditions and control of the physical and chemical properties of the compost. The utilization of biological passivators is widely acknowledged as the most cost-effective approach for heavy metal passivation; however, there remains a need to address how to enhance efficiency and simplify the process.

## 4. Mechanisms of Heavy Metal Passivation in Organic Solid Waste Composting

Due to the intricate composition of raw materials involved in composting and the complex physicochemical and biological changes occurring during the composting process, adequately investigating the passivation mechanism of heavy metals in current research poses challenges, and a comprehensive classification has not yet been established. We have categorized the mechanisms for heavy metal passivation in the composting process into ion exchange mechanisms, microbial community structure, degree of humification, and biomineralization mechanisms.

### 4.1. Ion Exchange Mechanisms

During the composting of organic solid waste, heavy metals in the bioavailable state (EXC and RED) may be reduced as the OM decomposes, and they can be adsorbed and immobilized by passivators through ion exchange mechanisms or by passivators providing cations to replace metal ions on the surface of the organic solid waste compost, thereby promoting co-precipitation with phosphate on the surface of ferromanganese oxides [51,56]. This passivation principle has been confirmed by a number of studies, and Soudejani et al. [57] showed that plagioclase has the ability to exchange alkali and alkaline earth metal cations with heavy metal ions and other cations (e.g., $NH^{4+}$) in the surrounding environment. Stylianou et al. [56] found that significant amounts of mobile heavy metals (carbonate-bound and exchangeable forms being predominant) could be captured from SS compost by a 20% (dry weight) addition of natural diabase zeolites and that the decomposition of OM through the composting process increased the formation of exchangeable forms of metals and contributed to the ion exchange of minerals. Wang et al. [51] studied the effect of Cu and Zn bioavailability during the co-composting of PM and RS and found that the passivation mechanism of composted heavy metals was very complex and involved multiple processes; however, they also found that the cations in the additive significantly exchanged $Cu^{2+}$ and $Zn^{2+}$ on the surface of the compost, thus promoting the formation of co-precipitation with phosphate on the surface of the iron and manganese oxides. Yang et al. [14] composted PM aerobically by adding green synthetic iron nanoparticles (G-nFe) and showed that the addition of G-nFe achieved efficient composting and the release of available phosphorus (AP), which was highly correlated with the passivation effect of Cd (Figure 1). The authors concluded that the mechanism of Cd passivation was through the precipitation of cadmium phosphate or co-precipitation with other phosphates.

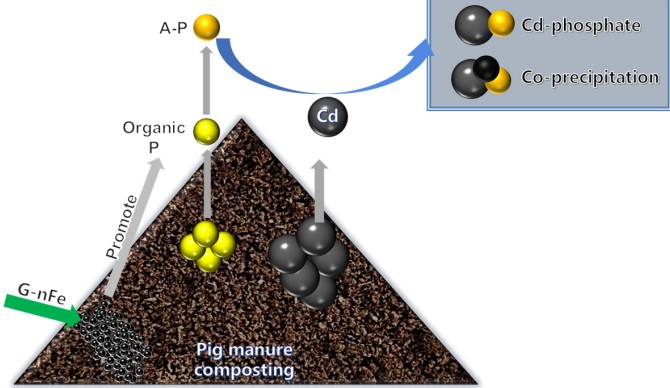

**Figure 1.** Mechanism of Cd passivation by the addition of G-nFe to pig manure aerobic compost.

### 4.2. Microbial Community Structure

Some passivators can regulate the transformation of carbon, nitrogen, and phosphorus and accelerate the decomposition of OM, improve the microbial community structure, and reduce the toxicity of heavy metals through direct microbial adsorption or enrichment/metabolism mechanisms [48]. In addition, microorganisms can regulate OM degradation and HS formation through the replication and expression of cellulase and ligninase genes and reduce the bioavailability of heavy metals through complexation [58]. This principle is discussed in Section 4.3. Zhang et al. [41] suggested that activated carbon affects the passivation efficiency of heavy metals by influencing environmental factors and changes in the bacterial community structure during the composting process. Moreover, there is a correlation between the various forms of different heavy metals and different species of microorganisms during the composting process. Cao et al. [59] evaluated the toxicity of heavy metals in the industrial-scale aerobic composting of livestock manure with Bacillus strains and found that the addition of the bacteria accelerated the degradation of OM in the compost to form FA and HA that were more readily bound to metal ions to form organically bound heavy metals, thereby reducing their biological effectiveness (Figure 2). The addition of Bacillus strains contributed to the passivation of heavy metals. Guo et al. [60] used structural equation modeling (SEM) to investigate the passivation pathways of heavy metals in compost with different C/N and different additives, revealing the complex passivation mechanisms of different heavy metals during the composting process. Hu et al. [46] studied the dynamics of bacterial communities and heavy metal composition during the composting of PM and maize straw and found that different bacteria played different roles for specific heavy metals, e.g., Bacteroidetes played a key role in the passivation of Zn, Cu, and Cr; Bacillus in Deinococcus-Thermus was involved in the translocation of Zn and Pb; and Bacillus deformans played a major role in the morphological transformation of the Cd fraction.

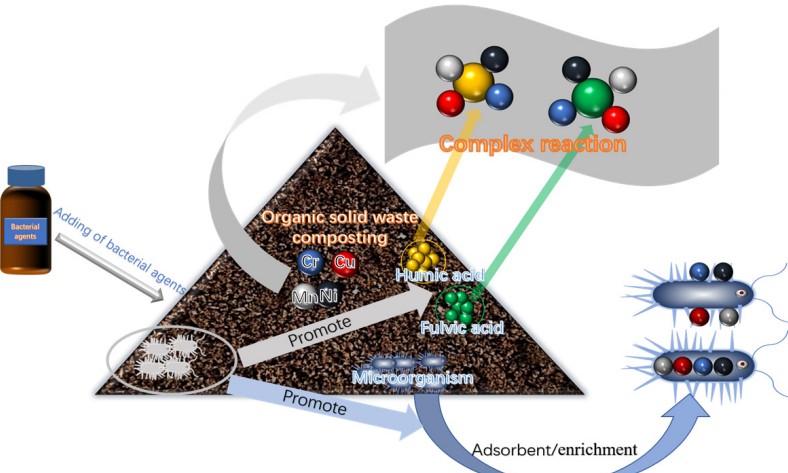

**Figure 2.** Mechanisms of heavy metal passivation by added bacterial agents during industrial-scale aerobic composting.

### 4.3. Degree of Humification

The HS content of compost is closely related to the presence of heavy metals. HS and HA are generally considered to contain a large number of surface functional groups such as carboxyl groups, phenolic hydroxyl groups, alcohol hydroxyl groups, amides, and aldehydes that combine with heavy metals to form binary or multiple complexes to reduce their mobility [61] (Figure 3). This passivation mechanism has been addressed using physical, chemical, and biological passivators, but the detailed mechanisms of HS migration and heavy metal transformation in compost are relatively poorly understood. Kong et al. [50] concluded that the passivation rate of heavy metals was significantly and positively correlated with HS, HA content, and the HA/anthranilic acid ratio and that

the addition of passivating agents to compost increased the passivation rate of heavy metals by promoting the humification process. The results showed that the higher the maturity and degree of humification of the compost, the more effective the passivation of heavy metals. Song et al. [11], in their study of biochar/montmorillonite-amended compost, found that organic components such as HS, HA, and FA, as well as dissolved organic carbon (DOC), were inextricably linked to morphological changes in Cu and Zn and that the use and conversion of low humic organic fractions in combination with enhanced humic components to promote potential heavy metal resistance/acting bacteria were the best way to improve heavy metal passivation rates. Liu et al. [42] investigated the effects of three combinations—phosphogypsum and calcium oxide (PPG + CaO), calcium superphosphate and calcium oxide (SSP + CaO), and zeolite—on compost maturity and heavy metal passivation in PM composting. The bioavailability of Cu, Cd, Pb, and Cr during composting could be reduced by 49.2%, 5.0%, 26.6%, and 54.3%, respectively, and the reductions were positively correlated with HA content and HA/FA.

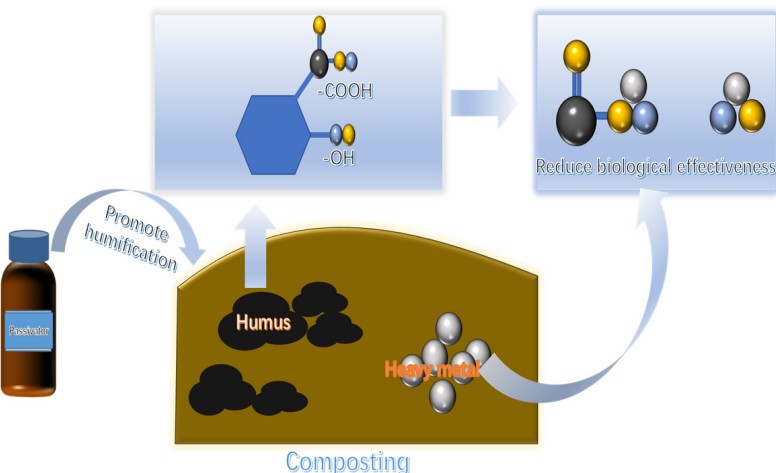

**Figure 3.** Mechanisms of the passivation effect on heavy metals by relevant groups produced during compost decay. Note: -COOH stands for carboxyl group and -OH stands for hydroxyl group.

### 4.4. Mechanisms of Biomineralization

Biomineralization of heavy metals refers to in situ bioremediation techniques that use microorganisms to participate in or accelerate the conversion of heavy metals to stable minerals. There are relatively few studies on the biomineralization of heavy metals in organic solid waste compost, but this phenomenon and the theory are important additions to compost heavy metal passivation and deserve further in-depth study. Chen et al. [62] investigated the effect of thermophilic bacteria on Pb passivation during sludge composting by inoculation with Streptococcus thermophilus FAFU013 and found that Pb(II) could be rapidly accumulated by selective biosorption and gradually converted to chlorophyll [$Pb_5(PO_4)_3Cl$] through biomineralization (Figure 4). By investigating the interaction between Pb(II) and bacterial strains isolated from Pb-Zn tailings, Chen et al. [63] found that Pb(II) immobilized by bacteria could be converted into rod-shaped Pb-hydroxyapatite [$Ca_{2.5}Pb_{7.5}(OH)_2(PO_4)_6$] nanocrystals and suggested that the synergistic effects of electrostatic attraction, ion exchange, and functional group chelation were the main factors contributing to the rapid biosorption of Pb. The biomineralization of heavy metals provides a new research idea in the field of heavy metal passivation in organic solid waste compost.

The passivation mechanism of heavy metals in compost is a complex process as it involves cation exchange and microbial community and humus changes during the composting process. Additionally, biomineralization also plays a role through cation exchange and humus participation. Therefore, multiple mechanisms are often involved in the passivation process of heavy metals in compost, and analyzing each mechanism's contribution to this process is crucial for understanding its complexity.

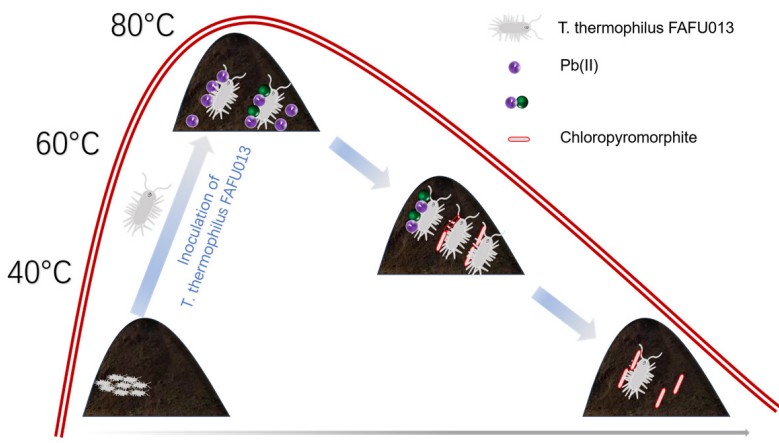

Figure 4. Biomineralization of Pb mediated by T. thermophilus during composting process.

**5. Conclusions**

In this study, we conducted an analysis of the current technologies utilized for the treatment of heavy metals in organic solid waste. The available studies have demonstrated that composting is one of the most well-established methods for treating organic solid waste. Furthermore, we performed a comprehensive review on the various types of passivators currently employed and categorized them into three distinct groups: physical, chemical, and biological. Our findings indicate that physical passivators are particularly advantageous due to their cost-effectiveness and the easy accessibility of raw materials while also offering flexibility in terms of composting conditions. Additionally, our investigation into the passivation mechanisms involved in composting revealed a diverse range rather than a singular process. We believe that, in the future, research should focus more on the interconnection of multiple passivation mechanisms and the development of composite passivation materials to improve the efficiency of heavy metal removal in organic solid waste compost. In addition, the cost-effectiveness and environmental benefits of passivation agents need to be taken into account in order to better target practical market applications.

**Author Contributions:** Y.Z.: Validation, data curation, original draft, and collating literature. W.Y.: Formal analysis, data curation, and funding acquisition. Q.Z.: Project administration and software. Z.C.: Review and editing. Q.C.: Resources, supervision, and review and editing. L.X.: Conceptualization, review and editing, supervision, and funding acquisition. All authors have read and agreed to the published version of the manuscript.

**Funding:** Natural Science Foundation of Fujian Province (No. 2020J01307), T; China is gratefully acknowledged.

**Institutional Review Board Statement:** Not applicable.

**Informed Consent Statement:** Informed consent was obtained from all subjects involved in the study. Written informed consent has been obtained from the participant(s) to publish this paper.

**Data Availability Statement:** Data available on request.

**Conflicts of Interest:** The authors declare no conflicts of interest.

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
