# Peer review of "Research Progress on Heavy Metal Passivators and Passivation Mechanisms of Organic Solid Waste Compost: A Review"

_fermentation, doi:10.3390/fermentation10020088_

Round 1
Reviewer 1 Report
Comments and Suggestions for Authors
An excellent and concise review. Some minor comments should be addressed to increase the quality of this paper.
1- Improve the quality of Manuscript text.
2- Use updated references to improve your conclusion.
3- Add the major gaps of current studies on the discussed topics in this paper.
4- Add DOI identifiers to all cited literature.
5- Make sure problematic papers such as retracted records were not cited within reference list.
6- Some figures can be merged. Please reorganize some of figures and improve the resolution of the given figures.
7- Add more details to figure captions.
8- In some parts of the manuscript text enough references have not been considered to discuss the text. Please carefully search the literature and re-discuss the given materials.
9- Please add research methodology to the paper. Please mention which databases were screened to evaluate the available literature on the discussed topic.
Reviewer 2 Report
Comments and Suggestions for Authors
The authors of the article deal with an interesting topic. the article is clearly and carefully prepared, the figures are colorful and creative. I think the article should be published. I have only one problem - the authors mostly quote the work of other Asian authors, there are more articles on this topic and from authors from other continents as well. Please complete and expand the article with relevant sources.
Comments on the Quality of English LanguageThe English language is at a good level, it is enough to check the spelling of some words.
Reviewer 3 Report
Comments and Suggestions for Authors
This is an review article. It analyzes the actual task of treatment compost contaminated with heavy metals. The article may be of interest to readers. The structure of the article is appropriate. It can be accepted for publication after minor revisions:
* Introduction. Need to mention how HM appears in organic solid waste? Proper organization of waste collection and management should not have HM.
* P.3. The authors write about "...mixing of industrial waste ...". Incorrect statement. Separate waste collection must be carried out. Solid organic waste cannot be mixed with hazardous waste.
* Chapter 3. There is a lack of analysis where various types of passivators are compared and the possibility of their optimal use is described.
* Chapter 4. After describing various HMs passivators mechanisms, it is necessary to analyze them more deeply: which ones are more important, what to pay primary attention to, how they how they affect each other, etc.
* Conclusions. Very general and not specific. The conclusions should emphasize the main results of the article (about passivators, about mechanisms, etc.).
